# Extreme–ultraviolet high–harmonic generation in liquids

Tran Trung Luu [1], Zhong Yin [1], Arohi Jain[1], Thomas Gaumnitz[1], Yoann Pertot[1], Jun Ma[1] & Hans Jakob Wörner[1]

High–harmonic generation (HHG) in gases has been the main enabling technology of attosecond science since its discovery. Recently, HHG from solids has been demonstrated, opening a lively area of research. In contrast, harmonic generation from liquids has so far remained restricted to low harmonics in the visible regime. Here, we report the observation and detailed characterization of extreme ultraviolet HHG from liquid water and several alcohols extending beyond 20 eV. This advance was enabled by the implementation of the recent liquid flat–microjet technology, which we show to facilitate the spatial separation of HHG from the bulk liquid and the surrounding gas phase. We observe striking differences between the HHG spectra of water and several alcohols. A comparison with a strongly–driven few–band model establishes the sensitivity of HHG to the electronic structure of liquids. Our results suggest liquid–phase high–harmonic spectroscopy as a new method for studying the electronic structure and ultrafast scattering processes in liquids.

[1] Laboratorium für Physikalische Chemie, ETH Zürich, 8093 Zürich, Switzerland. These authors contributed equally: Tran Trung Luu, Zhong Yin. Correspondence and requests for materials should be addressed to T.T.L. (email: trung.luu@phys.chem.ethz.ch) or to H.J.Wör. (email: hwoerner@ethz.ch)

Studies of electronic dynamics on their natural time scale have tremendously benefited from HHG in gases[1,2]. This technique not only provided a new source of coherent extreme ultraviolet (EUV) radiation but also enabled a variety of applications in spectroscopy and molecular imaging[3,4] using single or multiple attosecond pulses. Recently, HHG from solids[5–10] opened new possibilities for studying electronic dynamics[6,9,11–15] and electronic structure[7,8,16–18] in the bulk of solids. Extreme ultraviolet HHG from gases and solids has played a pivotal role in establishing and expanding real–time microscopic studies of electrons in matter. In the bulk of liquids, early efforts to observe HHG were limited to low–order harmonics in the visible domain[19]. EUV emission from water droplets has been observed[20], however it was restricted to incoherent plasma radiation. Coherent HHG could only be observed after laser-induced expansion of the droplets to densities lying 1–2 orders of magnitude below that of liquid water[20–22]. In an entirely different class of experiments[23], HHG was observed from the surface plasma and expanding high-pressure gas created by the interaction of an extremely intense laser pulse with a liquid jet. Therefore, EUV HHG from the bulk of liquids has not been reported so far.

In this work, we investigate the properties of EUV HHG from the bulk of liquids for both scientific and practical purposes using a new experimental approach: the liquid flat–microjet method (Fig. 1, see also refs.[24,25]). The flat microjet provides an ultrathin (~1.9 μm, see Methods), continuously renewed slab of liquid, which avoids the curvature of the interface inherent to spherical droplets and cylindrical microjets. These properties minimize the effect of reabsorption and phase mismatch, sample damage, as well as microfocussing of the infrared driver leading to divergence of the emitted harmonics, turning flat microjets into robust, reproducible and nearly flawless targets for HHG. This new approach enables us to isolate EUV high-harmonic emission from the bulk of liquids from that of the surrounding gas phase. Our detailed measurements moreover provide detailed information about the generation mechanism. We demonstrate a pronounced sensitivity of the HHG spectra of liquids to their electronic structure, especially to the widths of the bands and the band gaps. We further show that these spectral signatures are dominated by the microscopic non-linear response rather than by macroscopic propagation effects. Finally, we show that the ellipticity

dependence of HHG from liquids is considerably broadened compared to that of gas-phase HHG. These combined results suggest liquid-phase high-harmonic spectroscopy as a potential avenue for studying the electronic structure and sub-femtosecond electron scattering processes in liquids.

## Results

**HHG from liquids**. Figure 1a illustrates the experimental scheme. Intense infrared laser pulses are focused (f/45) onto a flat microjet made from two colliding cylindrical liquid microjets (~50 μm diameter) in high vacuum. The emitted spectra are recorded by a flat–field EUV spectrometer (see Methods). A key challenge, the separation of HHG from the liquid and the surrounding gas phase created by evaporation from the jet, is elegantly solved by making use of the wedge–like shape of the flat microjet, which converges from top to bottom[24]. This shape, combined with different refractive–index changes for the fundamental and high harmonics, results in a spatial separation of HHG from the bulk of liquids and the gas phase located behind the flat microjet (see Methods). We note that HHG from the gas phase located in front of the flat microjet is entirely absorbed by the jet because typical absorption lengths at our EUV energies are on the order of ~11 nm at 21 eV[26]. The simultaneously observed and spatially separated high-harmonic spectra from the liquid and gas phases are shown in Fig. 1c. The unique assignment of these spectra to the two distinct phases was further ascertained by scanning the position of the flat microjet in the direction perpendicular to the laser beam (see Methods).

We chose water and various alcohols as liquid samples because of their relevance to chemistry and biology. Their balanced viscosity and vapor pressure supports the generation of optimal flat microjets while preserving a sufficient vacuum to operate multichannel–plate–based detectors. Figure 1d shows the EUV emission observed from liquid ethanol as a sample. The spectrum consists of a series of odd harmonics, reaching up to H27 (of the 1.5 μm fundamental wavelength) and extending beyond 20 eV. Comparison to the simultaneously recorded HHG signal from gas–phase ethanol (Fig. 1c, bottom panel) shows remarkably different characteristics, i.e., much brighter emission at low energies and a lower cut–off for the liquid–phase emission.

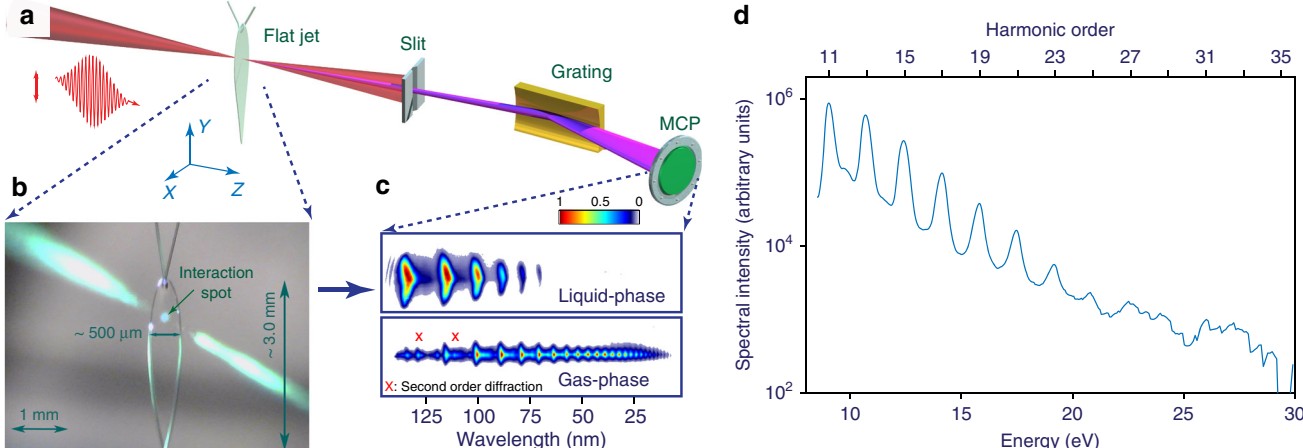

**Fig. 1** Extreme–ultraviolet high–order harmonic generation from liquid samples. **a** Experimental apparatus consisting of a liquid flat microjet and a flat–field spectrometer with a multi-channel plate (MCP). **b** Photograph of the laser–liquid interaction. The scattering of third harmonic (green) can be clearly observed. **c** Typical spatio–spectrally resolved far–field images of EUV emission from the liquid and gas-phase ethanol. The two spectra are individually normalized to the maximum peak intensities, as displayed in the color bar. **d** HHG spectrum recorded from interaction of laser pulses (30 fs, 1.5 μm, 1.2 mJ at 1 kHz, effective peak electric field strength of ≈1.5 V/Å inside the medium) with liquid ethanol

**Intensity scaling**. As a first step to elucidate the mechanism underlying liquid–phase HHG, we studied the high–harmonic spectral intensity dependence on the amplitude of the laser electric field. Figure 2a shows that the cut–off photon energy (or harmonic order) follows a nearly–linear dependence on the peak electric–field strength ($H_{\text{cut-off}} \propto E^{1.2}$). This linear dependence is similar to recent observations in solids[5] and contrasts with the quadratic scaling observed in gases[27]. We proceed to study the dependence of the intensity of individual harmonics on the electric–field strength. Figure 2b reveals a distinctively non–perturbative response of all observed harmonic orders (H13 to H21) for all field strengths. The scaling of H13 at the lowest field strengths lies closest to the perturbative response (blue dashed line in Fig. 2b), although it clearly deviates from it, whereas the scaling of H21 is far off the perturbative behavior (green dashed line).

**Coherence**. In the next step, we investigated the spatial coherence properties of HHG from liquids. Using a phase mask introduced into the driving laser beam, we created two spatially separated foci either inside the liquid or in the surrounding gas phase (Supplementary Note 1) and recorded the spatial interference fringes in the spatio–spectrally–resolved far–field images. This two–source spatial interferometry[28] is sensitive to the spatial and temporal coherence properties of the emission. Our observation of a nearly identical contrast of the liquid–phase and gas–phase emission suggests that the liquid–phase harmonics are no less coherent than those from the gas phase.

**Photon flux**. Having demonstrated the non–perturbative and highly coherent characteristics of EUV HHG from liquids, we furthermore determined its photon flux (see Supplementary Note 2). We measured a flux of $\approx 6 \times 10^6$ photons/laser shot in H11 at a driving pulse energy of $\approx 1.2$ mJ. In the present spectral range ($<20$ eV), the detected HHG originates mainly from the final $\approx 500$ nm of the sample (see Methods). Future technological advances, including the application of mid–infrared drivers, might significantly increase the cut–off photon energy. Furthermore, since the absorption length increases monotonically for e.g., water from its minimum of ~11 nm at 21 eV[26] to ~10 μm before the oxygen K–edge, extending the cut–off photon energy might considerably increase the photon flux of HHG from liquids.

**Effect of the nature of the liquid on HHG**. In the next step, we compared HHG spectra recorded from water and the most common alcohols under individually optimized experimental conditions (Fig. 3) to investigate their sensitivity to the nature of the liquid. Three characteristic features can be identified in their spectra. First, all spectra exhibit very clear odd–only HHG with cut–offs in the vicinity of 20 eV. Second, HHG spectra of all three alcohols decrease monotonically with photon energy. Third, the HHG spectrum of water shows a plateau (from H11 to H15), followed by a steep descent to the cut–off (from H17 to H21). While the first point is a standard feature for HHG in gases and inversion–symmetric or amorphous solids, the second point is remarkably distinct from HHG in gases or solids, whereas the third point is reminiscent of the characteristics of HHG from gases[1] and rare–gas solids[8].

**Numerical modeling**. With the goal of understanding the origin of these characteristic spectra, we now turn to numerical simulations. We confine our theoretical study to liquid water and ethanol since the electronic-structure properties of ethanol are similar to those of the other two alcohols[29], and so are their HHG

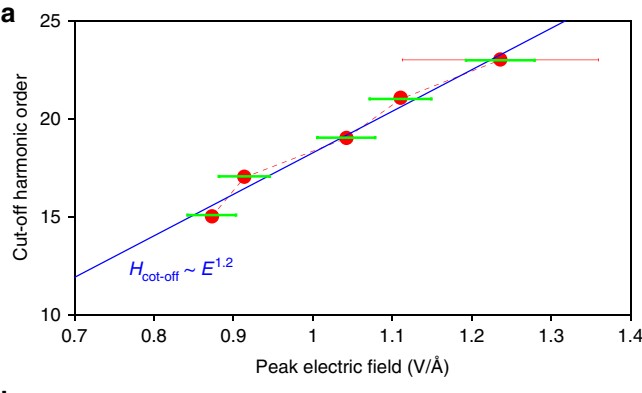

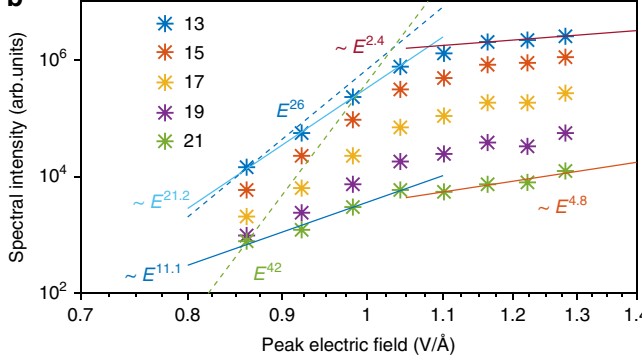

**Fig. 2** Non–perturbative high–order–harmonic generation from liquids. **a** Cut–off photon-energy dependence on peak electric field strength inside the medium (red dots). A typical absolute (total, approximated, propagated value of multiple measurements performed under identical experimental conditions) error bar of the electric field strength is displayed as a horizontal solid red line. Relative error bars are displayed as horizontal solid green lines. A linear fit of the high harmonic cut-off photon energy ($H_{\text{cut-off}}$) performed on a double-logarithmic scale is shown as the solid blue curve on a double-linear scale. **b** Dependence of the emitted intensity of individual harmonic orders on the peak-electric-field strength. A linear fit to the first four and the last three data points of H13 and H21 are shown on a double-logarithmic scale. Dashed lines represent the scaling law expected for a perturbative response. Liquid ethanol was used as a sample for all data shown here

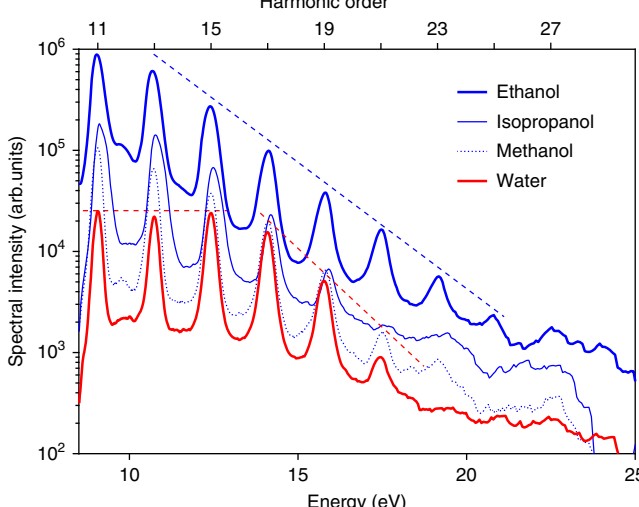

**Fig. 3** Comparison of HHG from liquid water and several alcohols. HHG spectra recorded from water and three alcohols under individually optimized experimental conditions. The dashed straight lines are guides to the eye

spectra. Although the electronic-structure properties of liquid water have been continuously explored for decades[30–39], performing a complete time–dependent Schrödinger equation calculation on liquid water remains unfeasible. We therefore opted for a computationally tractable approach based on the densities of states (DOS) extracted from state–of–the–art X–ray spectroscopy[40–42], see Fig. 4a. Specifically, a total of five model bands (three valence bands, VB1–VB3 and two conduction bands, CB1–CB2, Fig. 4b) were chosen and their energetic positions and curvatures were selected such that they reproduce the measured DOS, including the band gap. We note that the measured DOS of water and ethanol (Fig. 4a) display a strong resemblance, the main difference being the absence in the DOS of ethanol of the clear pre–peak assigned to the $4a_1$ band in the DOS of water[43,44]. This difference led us to use a broader conduction band (labeled CB1(E)) in the case of ethanol.

These bands were then used as input to quantum–mechanical calculations, i.e., the numerical solution of the semiconductor Bloch equations (SBE) in one dimension. In addition, to facilitate a direct comparison with the experimental spectra, we included an approximation of propagation and spatial integration across the laser beam profile (Methods). This theoretical approach is motivated by the following arguments (for more details, see Supplementary Note 3). First, the observed linear scaling of the cut–off energy with the driving field strength is reminiscent of the behavior previously observed in solids[5]. Second, it is generally accepted that water can be described as a very-large-band-gap semiconductor[45,46] on one hand, and that its electronic properties can be approximated by an effective band structure[47] due to the existence of local order on the other hand[43,48].

We first investigated the relative importance of the bands shown in Fig. 4b for explaining the observed spectra. We found that our simulations using five bands accurately reproduced the measured spectra. However, as we reduced the model to the minimum of two bands (highest valence band VB1 and lowest conduction band CB1 or CB1(E)), the spectra did not change significantly. The spectra obtained with this minimal 2–band model are shown in Fig. 4c. The calculated spectra (full lines) are in excellent agreement with the measured spectra (dashed lines). The three main spectral characteristics, i.e., the regular intensity

decrease in the ethanol spectrum, as well as the plateau in the water spectrum and even its spectral minimum at H13 (compared to H11 and H15), are all reproduced by our model calculations. As a consequence, we can relate the spectral differences in HHG from liquid water and ethanol to the different electronic structures of these two liquids.

A significant characteristic in the HHG spectrum of liquid water and absent for all alcohols studied, is the presence of a spectral minimum at H13. We investigated the importance of the curvatures of the conduction and valence bands by removing them, which is equivalent to converting the SBE to the optical Bloch equations. The disappearance of the dip at H13 for the case of flat bands (not shown) suggests the importance of these curvatures and therefore of intraband excitations in the HHG from liquid water. Furthermore, we found that the energetic position of the spectral minimum at H13 is also sensitive to the band gap as shown in Methods. We found that the minimum shifts linearly with the size of the band gap, therefore coinciding with the position of the odd harmonics when the band gap is modified by an even multiple of the fundamental photon energy.

These results show that the dominant spectral features observed in EUV HHG from liquids can be explained by the width of the conduction and valence bands, the band gap, and the peak electric field strength, in each case. We further verified the influence of macroscopic effects by solving the coupled Maxwell-SBE problem (Methods). The results of these calculations are in good agreement with both the experiment and the SBE results discussed above, showing that macroscopic effects have a minor influence on the observed spectral features.

**Probing the spatial extension of the electron–hole wave function.** Finally, we discuss the ellipticity dependence of HHG in liquids. We exploited our ability to simultaneously record HHG spectra of the liquid and gas phases to directly compare their ellipticity dependences, as shown in Fig. 5. The polarization of the incident laser pulses was varied from linear (ellipticity $\epsilon = 0$) to circular ($\epsilon = 1$) (Fig. 5a) without changing the pulse energy or duration. Interestingly, the ellipticity dependence of HHG from the liquid phase is broader compared to the gas phase. The width

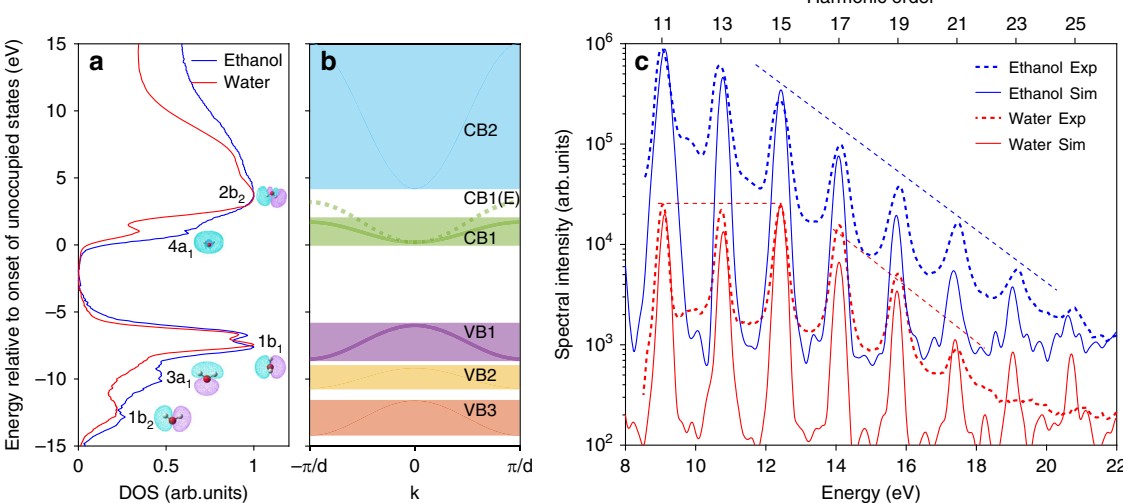

**Fig. 4** Sensitivity of HHG to the electronic structure of liquids. **a** Density of states extracted from X-ray absorption and emission spectra of liquid ethanol and water. The assignments of the bands to orbitals of isolated molecules are shown only for water. The orbitals are displayed as corresponding insets. **b** Adapted model band structure used in the numerical solution of the semiconductor Bloch equations. The modified first conduction band CB1(E) is used instead of CB1 in the case of ethanol. **c** Comparison of experimentally measured spectra (dashed curves) and calculated spectra using only the VB1 and CB1/CB1(E) (solid curves). Dashed straight lines are guides to the eye

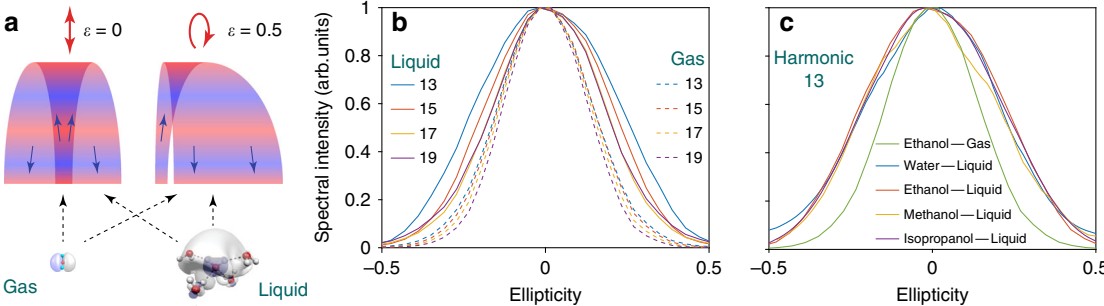

**Fig. 5** Ellipticity dependence of HHG from liquids. **a** Schematic illustration of valence orbitals of a water monomer and a water pentamer cluster (representative sub–unit of liquid water[50]) and the laser-driven continuum–electron wave packet for different ellipticities of the driving field. The color gradient illustrates the wave nature and propagation of the electronic wavepacket. $\epsilon$ is the ellipticity, with 0, 1 corresponding to linearly and circularly polarized electric fields, respectively. **b** Ellipticity dependence of the high–harmonic intensities, for different harmonic orders (H13 to H19) emitted from liquid–phase or gas–phase ethanol (solid and dashed lines, respectively) under identical experimental conditions. **c** Ellipticity dependence of H13 emitted from different liquids in comparison to gas–phase ethanol

of the measured ellipticity dependence of the gas phase agrees well with the typical ellipticity dependence of molecular HHG[49]. Figure 5c compares the measured ellipticity dependences of H13 for all four studied liquids with that of gas–phase ethanol. The ellipticity dependence of HHG from all liquids is nearly identical and clearly broader than the gas–phase reference.

Both our observations in liquids and the previous observation of a nearly unchanged ellipticity dependence of HHG from rare–gas solids compared to the isolated atoms[8] are consistent with the following qualitative interpretation. In the gas phase, the continuum–electron wave packet has to return to the parent ion to emit high harmonics, specifically to the spatial region defined by the ionized orbital(s) (see Fig. 5a). Therefore the width of the ellipticity dependence function is determined accordingly[49]. In the liquid phase, the valence–electron hole has a larger spatial extension, typically covering several molecules in the case of liquid water[50–52]. Therefore, high harmonics can still be emitted for larger ellipticities, as long as the returning continuum wave packet spatially overlaps with the extended valence hole. In a classical picture, this means that recombination can take place to a neighboring molecule. In the case of rare–gas solids, the valence–electron hole is expected to be spatially confined to a single atom, which is consistent with the observed small difference between gas–phase and solid–phase.

An additional effect that might contribute to a broadening of the ellipticity dependence in the liquid phase is electron scattering with neighboring molecules. These collisions would broaden the spatial extension of the continuum electron wave packet and cause partial decoherence, thereby suppressing the HHG amplitude. We estimate a maximal electron excursion length of 1.5 nm, i.e., a propagation length of 3 nm for the cut–off harmonic observed in our experiments (20 eV emitted in a 1.5 V/Å field). This is longer than most calculated elastic mean–free paths (EMFP) for liquid water[53,54], but shorter than, both, the EMFP (~13 nm) and the mean free path including all types of collisions (~4.0 nm), determined from experiments on amorphous ice[55] at the relevant kinetic energies (~ 11eV). In the absence of experimental data on liquid water at these energies, and assuming that electron transport in liquid water is similar to that in ice[55–57], we conclude that the number of collisions experienced by the strongly-driven electrons in our experiments amounts to one or less. Therefore, the spatial extension of the valence hole is likely to be the dominant effect in the observed broadening of the ellipticity dependence. Further extension of the theory described in ref. [58] may help in understanding this in more details.

## Discussion

Similar to the observation of EUV HHG from amorphous solids, our results may contribute to clarifying the role of short/long–range correlations in HHG from amorphous media. The EUV HHG from liquids observed in our experiments has a cutoff photon energy of about 20 eV, which is very similar to the maximum photon energy observed in fused silica[15,18]. However, it is much lower compared to the 35-eV cutoff photon energy of HHG from quartz[18], which possesses long–range order. Considering that the effective band gap of water is very similar to the band gap of fused silica (and quartz), this suggests that the limited cutoff photon energy observed in the present experiments might be due to the lack of long–range order.

In this work, we have demonstrated the generation and characterization of EUV high harmonics from liquids. This progress was enabled through the application of the innovative liquid flat–microjet technique, that elegantly separates high–harmonic emission from the liquid and gas phases. Intense, coherent HHG from four different liquids has been observed up to 20 eV, showing promise as an alternative source of coherent EUV radiation that combines the advantages of both gas and solid samples while alleviating their shortcomings (limited density and sample damage, respectively). We also demonstrated the sensitivity of HHG to the electronic structure of liquids, especially their density of states and their band gaps by comparison with a quantum–mechanical strongly–driven few–band model. These results establish the potential of liquid–phase high–harmonic spectroscopy to investigate the electronic structure of liquids. Finally, we reported a broadened ellipticity dependence of HHG from liquids compared to the same molecules in the gas phase. Such measurements might shed new light on the spatial extension of valence–electron holes in liquids and sub–femtosecond electron scattering dynamics in liquids.

## Methods

**Experimental apparatus**. We utilized a high-power Ti:Sapphire laser delivering 7 mJ, 1 kHz, 30 fs pulses at the carrier wavelength of 800 nm. The amplified laser pulses are sent to an optical parametric amplifier (HE-TOPAS-Prime, Light Conversion) to create signal pulses centered at 1500 nm with 1.2 mJ pulse energy. The signal pulses are focused onto the flat microjet inside a vacuum chamber with ƒ/45 to reach a peak-electric-field strength of ≈1.8 V/Å in vacuum as estimated from the calibration of the HHG cutoff in neon. The electric-field amplitude inside the liquid sample is then estimated using Fresnel's formula for S and P polarizations and assuming that a normal incident geometry is used. The incident angle due to the wegde-like shape of the flat microjet is negligible for this calculation. The fields inside the sample are calculated to be ~0.86 times the field strengths outside (neglecting interference effects due to the thin flat microjet) for all liquids due to their similar refractive indices. Under operating conditions, the flat–microjet

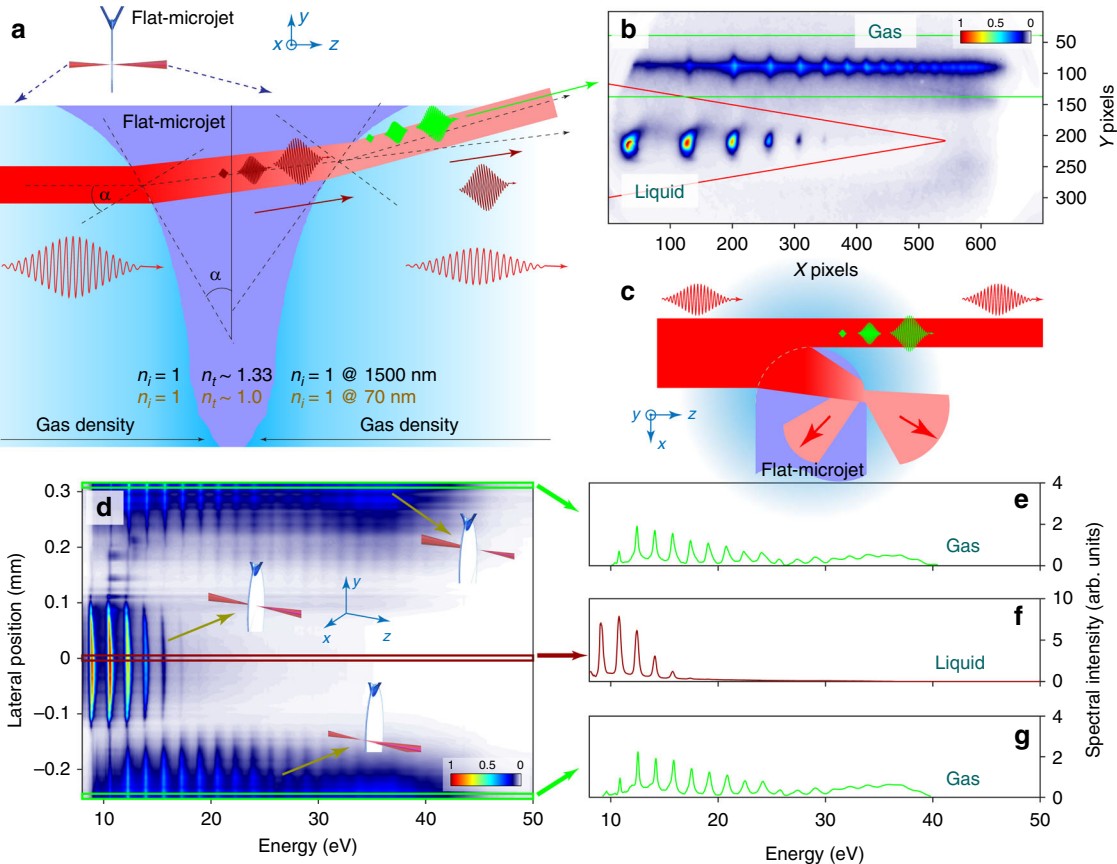

**Fig. 6** Separating HHG from liquid and gas. **a** Schematic illustration of the optical path of the infrared laser pulses through the liquid flat microjet. The EUV emission from the bulk liquid is not refracted as it leaves the jet because the change in refractive index $n_{i,t}$ (i: incident, t: transmitted) is negligible, in contrast to the driving infrared beam that is strongly refracted, resulting in **b**, spatial separation of the emissions from the liquid and gas phases. The red and green lines delineate the areas of integration. **c**, Illustration of the laser–beam–flat–microjet interaction at partial lateral overlap. **d** Typical HHG spectra integrated over the y-dimension of the detector, obtained by scanning the jet position in the x-dimension. Inset drawings illustrate where the laser beam hits the flat–microjet spatially. **e**–**g** Cross cuts of the HHG spectra when the laser beam hits the center of the flat microjet (**f**), or the surrounding gas phase on either side of the jet (**e**, **g**). Note the different scales on the vertical axes. Liquid ethanol is used as a sample in all data shown here

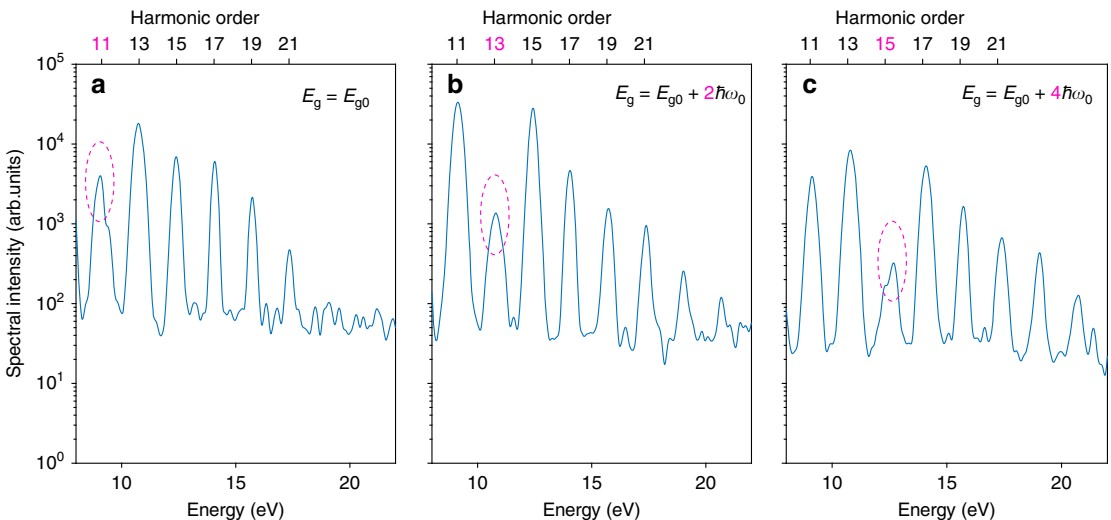

**Fig. 7** Sensitivity of HHG spectra to the band gap. **a**–**c**, Spectra calculated from the numerical solution of the SBE, using the water electronic structure with the band gap changed from $E_{g0} = 6.2$ eV to $E_{g0} + 2\hbar\omega_0$ and $E_{g0} + 4\hbar\omega_0$. Destructive interference resulting in a local minimum at harmonic 11, 13, 15, can be clearly observed. For clarity of the demonstration, spatial integration was turned off in these calculations and a weaker electric field strength of 0.7 V/Å has been used

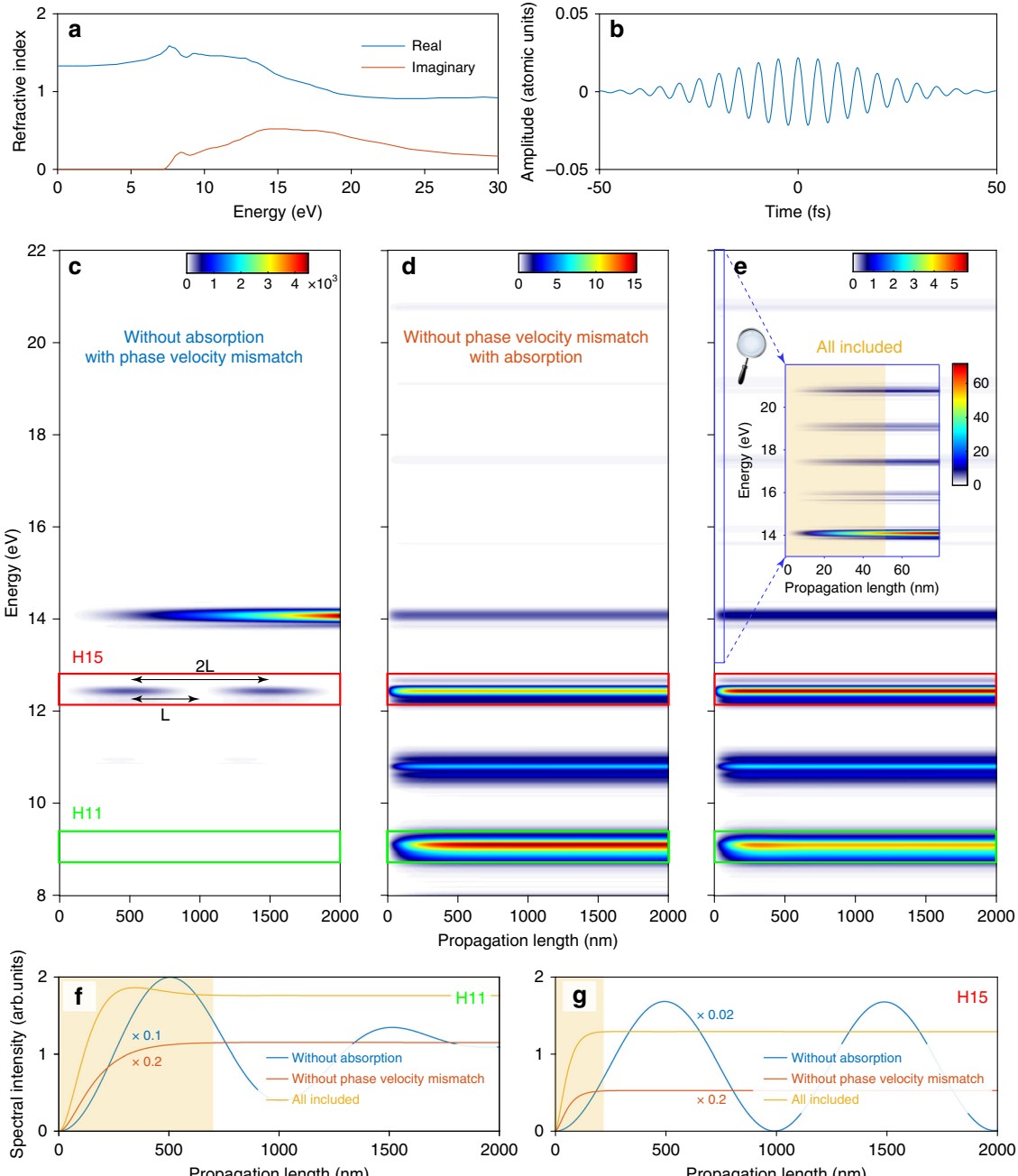

**Fig. 8** Propagation and phase-matching effects in liquid water obtained by solving the coupled Maxwell–semiconductor Bloch equations. **a** Complex refractive index[26] and **b**, electric field utilized in our simulations. **c–e** Spectral intensity build-up at different positions inside the liquid water flat microjet for three cases: **c**, keeping only the real part of the refractive index, while setting the imaginary part to zero; **d** keeping only the imaginary part of the refractive index while setting the real part to one; **e** using the complete complex refractive index. **f**, **g** Spectral intensity of H11 and H15 integrated over the vertical dimension of the rectangles in **c–e**. Their absolute intensities are scaled as indicated. The orange rectangles in **e–g** highlight the build-up length of the HHG intensity

chamber displays a pressure of ≈1 × 10⁻³ mbar maintained by two liquid-nitrogen cold traps, while the spectrometer has a pressure of ≈1 × 10⁻⁵ mbar.

The generated EUV spectra are recorded using a spectrometer consisting of a flat-field abberation-corrected grating (300 grooves/mm, Shimadzu), a multichannel-plate detector coupled to a phosphor screen, and a charge-coupled device camera. All spectra reported in the main text are corrected for grating efficiency, multichannel-plate response, and conversion from wavelength to frequency domain.

Flat–microjet apparatus: We developed our own flat-microjet system with inspiration from the commercially available system[24]. The liquid sample is pumped with a high-performance liquid chromatography (HPLC) pump through two cylindrical nozzles, creating two colliding microjets. For all experimental results presented in this paper two ≈50 μm nozzles have been used. We used white light

interferometry to measure the thickness of our flat microjet. The thickness is in the range of ~1.2–1.9 μm depending on the flow rate, type of liquid, and position of the measurement with respect to the flat microjet. The flow rate for the creation of a stable flat microjet in turn depends on the nozzle size and the viscosity of the sample, which is in the range of 2 ml/min for alcohols and up to 6 ml/min for liquid water. Under the chosen flow rates and nozzle sizes, the speed of the liquid exiting the nozzles is ≈10 m/s. With the laser focus size of ≈120 μm and the repetition rate of 1 kHz, this corresponds to complete sample renewal between each laser shot. Millipor-purified water with around 18 MΩ of electrical resitivity and 99.8% ethanol, 99.7% iso–propanol, 99.8% methanol have been used as liquid samples in our experiments.

It has been shown in ref.[59] that micro–droplets and small bubbles are produced upon interaction of intense laser pulses with liquids. The extreme energy of the

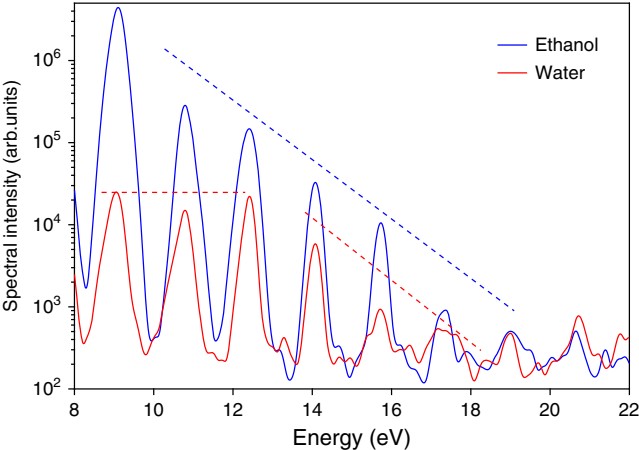

**Fig. 9** Weak dependence of HHG spectra on propagation effects. HHG spectra retrieved from coupled Maxwell-SBE calculations including propagation over the thickness of the jet using the band structures of water (solid red line) and ethanol (solid blue line) described in the main text. Dashed straight lines are guides to the eye. All characteristic features obtained from the SBE calculations (Fig. 4 of the main text) are preserved, including the uniform decrease in the ethanol HHG spectrum, as well as the plateau and the dip at H13 in the water HHG spectrum

laser pulse generates a vapor cloud immediately after hitting the jet. The vapor cloud then expands with hypersonic speed and forms a shock wave, the sound of which can be heard when the laser–flat–microjet interaction takes place under atmospheric pressure. In our experiments, although each laser shot sees a new spot of the liquid slab, the vapor cloud does not have enough time to escape completely after each laser shot and therefore accumulates. Therefore, the green area shown in the main text (Fig. 1b) and Supplementary Movie 1 is attributed to the scattering of the third harmonic (500 nm) on the vapor cloud generated by previous laser shots.

Distinguishing HHG from the liquid and gas phases: Due to the evaporation of molecules, the flat microjet is always surrounded by a gas phase from which high harmonics are also generated. We made use of the unique wedge-like shape of the flat microjet to spatially separate the emissions from gas and liquid phases. The detailed conceptual scheme is shown in Fig. 6a. Since the absorption length of liquids in this energy range is on the order of ~10 nm[26], HHG from gas located in front of the flat microjet will not be transmitted through the jet. Furthermore, because the refractive index of water at 18.5 eV is ≈1.0[26], the EUV components of HHG from liquids will not be refracted at the liquid-gas interface and thus propagate in a straight line to the spectrometer, in contrast to the emission from the gas phase generated by the transmitted laser pulse behind the liquid medium, which is deflected by an angle $\beta = \alpha - \sin^{-1}[\sin(\alpha) \times n_i/n_t] \approx \alpha(n_t - n_i)/n_t$, and is therefore spatially separated on the MCP detector. It has been shown[24] that at the top of the flat microjet, the typical curvature is on the order of $\alpha \approx 10$ mrad. Figure 6b shows the experimentally maximized vertical separation of the emissions observed on the MCP detector. The observed separation of ~7 mrad implies that our flat microjet presents a curvature of up to $\alpha \approx 30$ mrad in this demonstration.

A lateral scan of the flat microjet across the laser focus is used to further ascertain the assignment of the observed signals. A typical result of such a scan is shown in Fig. 6d where we vertically integrated the MCP images for different lateral positions. Integration over the ranges defined by the red and green lines isolates the HHG spectra of liquid and gas phases, respectively. The lateral scan further shows that the width of our flat microjet in this particular case is about 300 to 400 µm. At lateral positions of −150 and +150 µm relative to the center position, very weak HHG signal is recorded, which is explained in Fig. 6c. In this configuration, a small fraction of the laser beam will not interact with the liquid, thus HHG from the gas phase will be created. The remaining part of the laser beam is diffracted at the interface, then it is reflected internally or partially diffracted into a wide solid angle. Because of this complex diffraction and internal reflection, coherent build up of HHG cannot be realized and it will be inefficiently collected by the spectrometer because of the large solid angle of emission. As a consequence, this configuration yields very limited EUV flux from gas and almost no contribution from the liquid, explaining the observed suppression. Finally, the result of our lateral scan shows that the HHG from liquid is significantly stronger than HHG from the gas surrounding it as shown in Fig. 6e–g (note the different scales). Visual demonstration of these scans are presented in Supplementary Movies 1 and 2.

**Numerical simulations**. For all simulations shown in the main text, we used the well-established formalism of SBE[60–65] and its implementation to strongly-driven

condensed-matter systems[6,16,64,66]. The SBE are solved numerically for an initially unexcited few-band system in one dimension. The resultant spectra are shown in Fig. 4 of the main text. Propagation effects are included in first-order approximation using a previously described methodology[66]. The spatial integration is done as follows. Denoting $S(\omega) = S(\omega, r)$ as the microscopic spectral intensity generated at a given position $r$ in cylindrical coordinates, the spatially integrated spectral intensity on the detector $S_{avg}(\omega)$ can be calculated as:

$$S_{avg}(\omega) = \frac{1}{A} \int_0^{r_{max}} S(\omega, r) 2\pi r \mathrm{d}r, \qquad (1)$$

where $A$ is the area of the effective laser focal spot size of radius $r_{max}$. Evidently, the above integration strongly depends on the shape of the laser focal spot, which is assumed to be Gaussian as supported by our measured beam profile at the focus. Considering that liquids possess a density similar to or even higher than solids, electron scattering with nearest neighbors happens on similar time scales, which led us to choose a dephasing time $T_2 \approx 1$ fs[66]. Finally, we have neglected Coulomb interactions in our calculations because they have been shown to play a significant role only in the weak–field regime[63].

In the next step, we demonstrate the effect of the bandgap on the position of the intensity minimum and the shape of the HHG spectrum of water. The results of the simulations are shown in Fig. 7. In all of these simulations, the parameters are kept constant, except for the bandgap $E_g$ which is changed gradually from the starting value $E_{g0}$ with a step size of $2\hbar\omega_0$, where $\omega_0$ is the angular frequency of the carrier wave. For clarity of the main physical message, we turned off spatial integration and reduced the peak electric-field strength. The results demonstrate that the position of the intensity minimum shifts linearly with the size of the bandgap. Moreover, the position of the dip also depends on the intensity which is an expected behavior in strongly-driven systems. However, this effect is not apparent when the CB1(E) is used instead, suggesting that the formation of the plateau in HHG from water is collectively due to the size of the bandgap and the characteristic density of states of water.

**Phase matching in liquid water**. In the following, we investigate the effects of absorption and phase-velocity mismatch (i.e., phase matching) on HHG from liquid water. Our propagation formalism is based on the slowly-varying-envelope approximation[67]. This combination of the SBE as the microscopic nonlinear response and the first-order wave propagator in the frequency domain represents the state-of-the-art numerical treatment of HHG. This approach has been traditionally applied to gases, and its application to solids was suggested[61], but to the best of our knowledge, it has not been performed for HHG from solids or liquids yet. The results of the simulation are shown in Figs. 8, 9. Taking into account the accurately measured complex refractive index of liquid water[26,68], we can unambiguously quantify multiple macroscopic effects.

Phase–velocity mismatch: Since the imaginary part of the refractive index is associated with the absorption and the real part with the phase-velocity mismatch, we isolated the effect of phase-velocity mismatch by artificially turning off the absorption (setting the imaginary part to zero) before starting the propagation calculation. The results are shown in Fig. 8c. In the absence of absorption the total generated flux is very high as seen from the amplitude of the color map. Moreover, there is a difference between the real part of the refractive index at the fundamental wavelength and high-order-harmonic wavelengths, the generated HHG and the fundamental will be out of phase after some distance ($L$) of propagation and then come back in phase after a longer distance ($2L$). This effect manifests itself as an oscillatory behavior of the intensities of H11 and H15 as shown in Fig. 8f, g.

Absorption: We performed the complementary analysis by keeping the imaginary part unchanged while setting the real part to one. This keeps the absorption while artificially turning off any possible phase-velocity mismatch. The results are shown in Fig. 8d. In the absence of phase-velocity mismatch, there is no oscillatory behavior of the high-harmonic intensities. The absorption process slowly competes with the generation of high harmonics, resulting in saturated build–up curves (orange lines in Fig. 8f, g). In this case, the asymptotic flux is much weaker due to the absorption of photons.

By keeping the entire complex refractive index unchanged, our propagation simulations include the generation of new photons, the loss of photons due to absorption of liquid water, and the modulation of the photon flux caused by phase matching during propagation. The results are shown in Fig. 8e. Essentially, all generated harmonics reach saturation quickly after propagating inside liquid water (yellow curves in Fig. 8f, g). The coherent build-up length of the H11 photons can be above 500 nm while for higher-order harmonics, it can be ~50 nm or even shorter for very high energy photons. These characteristic length scales can be understood as the information depths over which information on electronic structure and electronic dynamics inside liquids can be obtained.

Furthermore, our coupled Maxwell-SBE simulations also reveal the dominant role of the microscopic emission in shaping the final spectra. The effects due to propagation inside liquids are less important. The results are shown in Fig. 9.

## Data availability

The data that support the findings of this study are available from the corresponding authors upon reasonable request.

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

## Acknowledgements

It is our pleasure to thank Andreas Schneider, Mario Seiler, Andres Laso, Markus Kerellaj, Kristina Zinchenko, Martin Huppert, Yulia Pushkar and Adam Smith for their contributions to the construction of the experiment. We gratefully acknowledge funding from an ERC Starting Grant (307270–ATTOSCOPE), an ERC Consolidator Grant (772797-ATTOLIQ), the ETH Zurich Postdoctoral Fellowship Program (FEL–31 15–2), the Marie Curie Actions for People COFUND Program, and SNSF R'equip grant 206021_170775.

## Author contributions

H.J.W. proposed the experiment. T.T.L and Z.Y. combined the flat-jet apparatus with the XUV spectrometer, carried out the experiments, performed the analyses and calculations and measured the thickness of the flat microjet. A.J., T.G., Y.P., and J.M. contributed to the development of the flat–microjet assembly, which was finalized and tested by A.J. and T.G. All authors discussed the results, T.T.L., Z.Y., and H.J.W. wrote the manuscript with input from all authors.

## Additional information

**Competing interests:** The authors declare no competing interests.

