## [Peer Review File · Nature Communications]

Reviewers' comments:

Reviewer #2 (Remarks to the Author):

I am satisfied generally with authors' effort in addressing my comments and questions. Perhaps there were some misunderstanding, and authors might have over-responded, which I outline below and they should be able to easily address them in the revision. Mainly, it has been clear that liquid media is not another promising light source (at least not yet) but this manuscript opens up an interesting avenue to study electronic structure and ultrafast scattering process in liquid phase. I suggest authors to work on the first paragraph of the introduction and make that message clear. As of right now it reads as an alternate light source, and when readers read the full paper they maybe disappointed because of the low flux and low cutoff from liquid compared to gas. This is probably why reviewer #1 is unsatisfied, which is understandable.

One approach, although this is completely up to the authors, could be that they can say what high-harmonics in gas gave us (imaging structure of molecules, attosecond pulses...), then say what solid phase experiments taught us (role of valence, conduction band, linear scaling, Berry's curvature etc.), then open up a discussion about how can we measure electronic structure of liquid and what is the role of relevant parameters such as periodicity (perhaps by mentioning the work on amorphous glass) then show these exciting results....

About potential misunderstanding: I am aware that both amorphous solids and liquid medium exhibit local correlations, such as interatomic distances, for example distance between neighboring oxygen atoms in liquid water. Here, I really meant the medium or long range periodicity. I still think that authors have to do a better job on addressing the role of short, medium and long range correlations to high harmonic generation?

For example, they can make use of published previous work on isolated atoms in a gas, perfectly periodic system such as single crystal quartz or zinc oxide, amorphous medium such as glass (should reference this paper: High-harmonic generation in amorphous solids, Nature communications 8, 724, 2017), and now liquid medium such as water. Given the availability of a wide range of experimental data, including liquid medium now, authors should take this opportunity.

A minor point. Manuscript reads "Finally, our SBE calculations also support this picture because they indicate that HHG in liquid water and alcohols is entirely dominated by interband polarization, suggesting the applicability of the generalized re-collision picture [45]." What is authors view on the possibility of re-collision to the neighboring site? (see. Phys. Rev. X 7, 021017, and Nature physics 13, 345-349 (2017)).

If authors agree with above I strongly recommend the paper for publication.

Reviewer #3 (Remarks to the Author):

Review: Extreme-ultraviolet high-harmonic generation in liquids by Tran Trung Luu, Zhong Yin, Arohi Jain, Thomas Gaumnitz, Yoann Pertot, Jun Ma, and Hans Jakob Wörner

I am still not convinced that this is the first study to show HHG from liquids such as water. In their reply they briefly discussed the work of Heissler et al. – a work that shows HHG generation with "real" liquid beams. I think that a couple of researchers tried water beams as targets for HHG and basically gave it up due to performance reasons. Since liquid droplets are not too far away from liquid beam targets the following publication may be another example for HHG generation (I think it is mentioned in the paper though).

A. Flettner, T. Pfeifer, D. Walter, C. Winterfeldt, C. Spielmann, and G. Gerber, High-harmonic generation and plasma radiation from water microdroplets, *Appl. Phys. B* 77, 747 (2003).

So, I still not believe that this is the first ever publication on HHG generation in liquids or liquid targets.

The authors argue simply "The lack of high-harmonic emission from water droplets has even been interpreted as the impossibility of generating extreme-ultraviolet (EUV) radiation from water at its liquid density. „What is the difference here – just the flat jet? By the way, the flat jet is a commercial system, where the "source" has to be mentioned. It does not seem to be an own development.

Since the performance is not very good I believe that the following may be an exaggeration: "Our work also establishes liquid flat microjets as a reproducible source of bright, coherent EUV radiation". Yes, it produces HHGs but with limited performance.

I appreciate though that the authors want to use this technique to study electronic states of molecules in liquids. The idea is interesting, in principle, but it may just be too early to state „liquid-phase high-harmonic spectroscopy as a promising new method for investigating the electronic structure and dynamics of liquids" based upon the given results. In summary, the results are not very specific and future will tell if they are competitive with other techniques.

The revised version of the paper is certainly interesting and deserves to be published, but I have some doubts that this is the ground breaking news the readers of NatureX (Nature Communications) would like to read. It certainly requires some revisions and the authors have to take out some overstatements and very general statements that may not hold.

I believe that the current manuscript should be published in a more specialized journal of *J. Phys. Chem. Lett.*. Maybe it is suitable for *Sci. Rep.* too.

Response to the referees

We would like to thank the two referees for their valuable time that they have taken to review our manuscript a second time. We particularly thank them for their excellent suggestions, which helped us to further improve our manuscript. To facilitate their reading, we have highlighted the additions/changes (abstract and main text) in our resubmission. We hope that the referees will find our answers and the revised manuscript satisfactory.

Referee #2

I am satisfied generally with authors' effort in addressing my comments and questions. Perhaps there were some misunderstanding, and authors might have over-responded, which I outline below and they should be able to easily address them in the revision. Mainly, it has been clear that liquid media is not another promising light source (at least not yet) but this manuscript opens up an interesting avenue to study electronic structure and ultrafast scattering process in liquid phase. I suggest authors to work on the first paragraph of the introduction and make that message clear. As of right now it reads as an alternate light source, and when readers read the full paper they maybe disappointed because of the low flux and low cutoff from liquid compared to gas. This is probably why reviewer #1 is unsatisfied, which is understandable.

One approach, although this is completely up to the authors, could be that they can say what high-harmonics in gas gave us (imaging structure of molecules, attosecond pulses...), then say what solid phase experiments taught us (role of valance, conduction band, linear scaling, Berry's curvature etc.), then open up a discussion about how can we measure electronic structure of liquid and what is the role of relevant parameters such as periodicity (perhaps by mentioning the work on amorphous glass) then show these exciting results....

We thank the referee for these excellent comments and suggestions. We have rewritten the end of our abstract and the complete introductory paragraph following these suggestions.

About potential misunderstanding: I am aware that both amorphous solids and liquid medium exhibit local correlations, such as interatomic distances, for example distance between neighboring oxygen atoms in liquid water. Here, I really meant the medium or long range periodicity. I still think that authors have to do a better job on addressing the role of short, medium and long range correlations to high harmonic generation?

For example, they can make use of published previous work on isolated atoms in a gas, perfectly periodic system such as single crystal quartz or zinc oxide, amorphous medium such as glass (should reference this paper: High-harmonic generation in amorphous solids, Nature communications 8, 724, 2017), and now liquid medium such as water. Given the availability of a wide range of experimental data, including liquid medium now, authors should take this opportunity.

A minor point. Manuscript reads "Finally, our SBE calculations also support this picture because they indicate that HHG in liquid water and alcohols is entirely dominated by interband polarization, suggesting the applicability of the generalized re-collision picture [45]." What is

authors view on the possibility of re-collision to the neighboring site? (see. Phys. Rev. X 7, 021017, and Nature physics 13, 345-349 (2017)).

We thank the referee for these nice suggestions. We have added one paragraph and two sentences in the main text to elaborate on our understanding of the role of short/long range correlations as well as the possibility of re-collision to the neighboring sites. All references suggested by the referee have been included.

If authors agree with above I strongly recommend the paper for publication.

We agree with all comments of the referee and have followed his/her suggestions. We thank the referee for this recommendation.

Referee #3

I am still not convinced that this is the first study to show HHG from liquids such as water. In their reply they briefly discussed the work of Heissler et al. – a work that shows HHG generation with “real“ liquid beams.

We thank the referee for these valuable points and are pleased to address them in detail.

Since liquid droplets are not too far away from liquid beam targets the following publication may be another example for HHG generation (I think it is mentioned in the paper though).

A. Flettner, T. Pfeifer, D. Walter, C. Winterfeldt, C. Spielmann, and G. Gerber, High-harmonic generation and plasma radiation from water microdroplets, Appl. Phys. B 77, 747 (2003).

So, I still not believe that this is the first ever publication on HHG generation in liquids or liquid targets.

We thank the referee for pointing out this reference, which was indeed cited in our submission (old Ref. [9], new Ref. [20]) and discussed, but perhaps too briefly. In the work of Flettner et al., only incoherent EUV emission (attributed to plasma luminescence) was observed from the droplets at their native density (that of liquid water, $\sim 3 \times 10^{22} \text{ cm}^{-3}$). For this reason, Flettner et al. used a pump-probe scheme, in which the pump pulse served to induce an expansion of the water droplets. The time-delayed probe pulse was then used to generate the EUV radiation. Coherent high-harmonic generation was only observed for pump-probe delays longer than one to a few nanoseconds, i.e. at time delays at which the density of the droplets had decreased well below that of liquid water. Subsequent work by H. Kurz et al. (PRA 87, 063811 (2013) and PRX 6, 031029 (2016)) used a model to estimate the corresponding maximal tolerable target densities, which were found to lie in the range of 10^{21} cm^{-3} (see Fig. 1 in PRX 6, 031029 (2016)).

Therefore, all three previous publications on HHG from droplets agree with each other in stating that coherent EUV HHG is not possible at the density of liquid water. This conclusion stands in contrast to our results, which were obtained with an entirely different target geometry (the flat jet). The spatial separation of high-harmonic emission from the liquid and gas phases observed in our experiments (see Fig. 1), as well as their different spectral shapes, prove that our experiments are indeed probing high-harmonic emission from the bulk of the liquid flat jet.

Last but not least, the other major difference between our experiment and the works on HHG from surface (new Ref. 23, and PRL, 98, 103902, (2007), Nature Physics, 2, 456-459, (2007), etc.) is that we detected the emission of EUV photons in the transmission geometry, whereas all

of these experiments utilized reflection geometry. That confirmed the nature of our EUV photons coming from the bulk of liquids, rather than from surfaces or interfaces.

The authors argue simply “The lack of high-harmonic emission from water droplets has even been interpreted as the impossibility of generating extreme-ultraviolet (EUV) radiation from water at its liquid density. „What is the difference here – just the flat jet?

We thank the referee for requesting this clarification. The main difference between the works of Flettner et al., Kurz et al. and our work, is that our experiments demonstrate coherent high-harmonic generation from bulk liquid water (and other liquid alcohols) at their native density. This was not the case in the previous works, as we explained above.

The main technical difference between the two approaches is indeed the flat jet, which differs from the previously used droplets by offering an ultrathin flat surface, which avoids the spherical geometry inherent to droplets. In comparison to cylindrical jets, the flat jet also avoids the curvature caused by the cylindrical geometry. In both cases, the curvature of the medium acts on the incident infrared beam as a lens, resulting in micro focusing of the radiation and, correspondingly, of the generated EUV radiation, which will therefore diverge after exiting the medium.

Fig. 1: Electric-field profile of an 800-nm laser pulse propagating from bottom to top, incident on a cylindrical liquid-water microjet with a diameter of 4 μm. A strong focusing of the infrared radiation by the microjet is observed, leading to a focus position located behind the jet. A similar, ray tracing version of this was already shown in our SI, Fig. S1c.

A flat liquid medium entirely avoids this micro focusing effect and therefore enables the generation of a weakly diverging EUV beam.

We propose this effect as the explanation for the different observations made on water droplets and the flat microjets. In other words, we believe that the medium curvature was one of the main factors that prevented the observation of coherent high-harmonic emission in previous experiments.

Furthermore, the ultrathin thickness of the flat jet does not suppress the incident electric field strongly. Thus it enables microscopic studies, given the fact that macroscopic effects are not dominant, as proven by our Maxwell-SBE simulations.

By the way, the flat jet is a commercial system, where the “source“ has to be mentioned. It does not seem to be an own development.

We thank the referee for pointing this out. We would like to mention that our flat-jet apparatus is an in-house development following the principle of the setup reported in Ekimova et al. Struc.

Dyn. 2015 (old Ref [15], new Ref [24]). We attach here a picture of our flat-jet set-up in Fig 2. Furthermore, we add a sentence in the SI 1A that our flat jet design follows the commercial apparatus.

Fig. 2: In-house developed flat jet apparatus following Ekimova et al. Struc. Dyn. 2015.

Since the performance is not very good I believe that the following may be an exaggeration: “Our work also establishes liquid flat microjets as a reproducible source of bright, coherent EUV radiation“. Yes, it produces HHGs but with limited performance. I appreciate though that the authors want to use this technique to study electronic states of molecules in liquids. The idea is interesting, in principle, but it may just be too early to state „liquid–phase high–harmonic spectroscopy as a promising new method for investigating the electronic structure and dynamics of liquids“ based upon the given results.

To soften our claim we have changed the abstract, removed/replaced the sentences and phrases accordingly. Please see our list of changes (end of this text) for details.

In summary, the results are not very specific and future will tell if they are competitive with other techniques.

The revised version of the paper is certainly interesting and deserves to be published, but I have some doubts that this is the ground breaking news the readers of NatureX (Nature Communications) would like to read. It certainly requires some revisions and the authors have to take out some overstatements and very general statements that may not hold.

We thank the referee for concluding that our manuscript deserves publication. We have removed the overstatements and very general statements that might have been disturbing in the previous version. We hope that our arguments given above have convinced the referee that our work indeed represents the first observation of coherent EUV high-harmonic generation from real bulk liquids, instead of expanded droplets with densities corresponding to high-pressure gases. Since this demonstration makes a new phase of matter available for EUV high-harmonic generation, we hope that the referee will support publication in Nature Communications.

In addition, we would like to point out that our results are not limited to the proof-of-principle demonstration of EUV HHG from liquids. We have included a considerable amount of information, such as the cut-off scaling, the intensity scaling, the comparison of 4 different liquids, theoretical modeling including propagation calculations and the ellipticity dependence. This combined information provides a detailed picture of the properties of EUV HHG from liquids, which we expect to be of interest to both experimentalists and theoreticians working in the field of ultrafast science.

Finally, our work shows that HHG spectroscopy of liquid samples has potential for accessing the electronic structure of liquids and ultrafast scattering dynamics. In both respects, the technique offers important advantages compared to other methods. For example, high-harmonic spectroscopy offers sub-femtosecond time resolution through the unique mapping from electron-transit time to the emitted photon energy in short electron trajectories. This aspect might be exploited in future experiments to resolve electron scattering dynamics or ultrafast structural dynamics on these time scales. As a second example, high-harmonic spectroscopy is primarily bulk sensitive (as shown in the build-up profiles of Fig. S6), as opposed to soft X-ray photoelectron spectroscopy, which is primarily surface sensitive. Finally, since the liquid phase is the most important phase of matter for chemical and biological processes, liquid-phase HHG spectroscopy may open new research directions of relevance to chemistry and biology. For all these reasons, we believe that Nature Communications would be an appropriate journal for publishing our results.

List of (major) changes:

- We deleted the second sentence in our abstract, rewrote the last sentence, removing the claim of the EUV radiation as a bright source, to soften the claims as suggested by referee #3.
- In the abstract, we replaced the word “breakthrough” by “advance”, to also soften our claim.
- We rewrote completely the introductory paragraph, as suggested by referee #2. The main point is shifted more towards spectroscopy.
- Page 2, second paragraph, we added in a phrase to emphasize how our flat-jet overcomes geometrical limitations of micro droplets or cylindrical microjets.
- Page 3, first sentence: we added a sentence hinting at how our results may help in studying electronic structure and ultrafast scattering processes in liquids, as suggested by referee #2.
- Page 5, final paragraph: we changed the title from “Different liquids” to “Effect of the nature of the liquid on HHG”
- Page 10, second to last paragraph: we added in completely one paragraph describing our opinion on the influence of short/long range order on HHG from liquids, as suggested by referee #2.
- Page 11, first paragraph, first sentence: we removed the claim that our results “represented a paradigm shifted compared to previous works”, to soften the claims, as suggested by referee #3.
- SI page 2, 1A, first sentence: we added the following: We developed our own flat-microjet system with inspiration from the commercially available system [1].

REVIEWERS' COMMENTS:

Reviewer #2 (Remarks to the Author):

Authors have adequately addressed my concerns. Revised manuscript introduces the experiment in a way that it might be presenting a new platform for high harmonic spectroscopy in liquid media as opposed to presenting another bright (actually not so bright) XUV source. Discussion with respect to amorphous systems seem appropriate.

I also think that the revised manuscript and reply addresses concerns of other reviewer particularly with respect to Flattener's work published in Appl. Phys. B 77, 747 (2003). In that paper the generation process is described using re-collision model assuming that after 600 picosecond the interatomic distances are longer than the electron excursion distances. Here, although not confirmed through time domain measurements, the understanding of authors is that liquid is more or less intact during the time when high-harmonics are produced, which is basically within the duration of laser pulse. Therefore, I think this experiment is conceptually different from Flattener's work although there are technical similarities such as using micro-droplets.

Response to the referees

We would like to thank referee #2 for his/her valuable time that he/she has taken to review our manuscript a third time.

Referee #2

Authors have adequately addressed my concerns. Revised manuscript introduces the experiment in a way that it might be presenting a new platform for high harmonic spectroscopy in liquid media as opposed to presenting another bright (actually not so bright) XUV source. Discussion with respect to amorphous systems seem appropriate.

I also think that the revised manuscript and reply addresses concerns of other reviewer particularly with respect to Flattener's work published in Appl. Phys. B 77, 747 (2003). In that paper the generation process is described using re-collision model assuming that after 600 picosecond the interatomic distances are longer than the electron excursion distances. Here, although not confirmed through time domain measurements, the understanding of authors is that liquid is more or less intact during the time when high-harmonics are produced, which is basically within the duration of laser pulse. Therefore, I think this experiment is conceptually different from Flattener's work although there are technical similarities such as using micro-droplets.

Thank you for your great comments regarding our revision, especially with the revised introduction and the added discussion with respect to HHG from amorphous systems. Thank you for your confirmation that our work is conceptually different from Flettner's work. Furthermore, we would like to clarify that our work used a flat sheet provided by the flat micro-jet, and not micro droplets, therefore our work is also in this technical regard a different approach.

Once again, thank you so much for your great support. We appreciate it very much!